# Contrastive Learning with Stronger Augmentations

## Abstract

Representation learning has been greatly improved with the advance of contrastive learning methods with the performance being closer to their supervised learning counterparts. Those methods have greatly benefited from various data augmentations that are carefully designated to maintain their identities so that the images transformed from the same instance can still be retrieved. Although stronger augmentations could expose novel patterns of representations to improve their generalizability, directly using stronger augmentations in instance discrimination-based contrastive learning may even deteriorate the performance, because the distortions induced from the stronger augmentations could ridiculously change the image structures and thus the transformed images can not be viewed as the same as the original ones any more. Additional efforts are needed for us to explore the role of the stronger augmentations in further pushing the performance of unsupervised learning to the fully supervised upper bound. Instead of applying the stronger augmentations directly to minimize the contrastive loss, we propose to minimize the distribution divergence between the weakly and strongly augmented images over the representation bank to supervise the retrieval of strongly augmented queries from a pool of candidates. This avoids an overoptimistic assumption that could overfit the strongly augmented queries containing distorted visual structures into the positive targets in the representation bank, while still being able to distinguish them from the negative samples by leveraging the distributions of weakly augmented counterparts. The proposed method achieves top-1 accuracy of 76.2% on ImageNet with a standard ResNet-50 architecture with a single-layer classifier fine-tuned. This is almost the same as 76.5% of top-1 accuracy with a fully supervised ResNet-50. Moreover, it outperforms the previous self-supervised and supervised methods on both the transfer learning and object detection tasks.

## 1 Introduction

Deep neural network has shown its sweeping successes in learning from large-scale labeled datasets like ImageNet (Deng et al. (2009)). However, such successes hinge on the availability of a large amount of labeled examples that are expensive to collect. To address this challenge, unsupervised visual representation learning and self-supervised learning, have been studied to learn feature representations without labels. Among them is the contrastive learning (Hadsell et al. (2006); Misra & Maaten (2020); Chen et al. (2020b); He et al. (2020); Caron et al. (2020)), showing great potentials to close the performance gap with supervised methods.

In contrastive learning Hadsell et al. (2006), each image is considered as an instance, and we wish to train the network so that the representations of different augmentations of the same instance are as close as possible to each other (He et al. (2020); Chen et al. (2020a); Wu et al. (2018); Hjelm et al. (2018); Oord et al. (2018); Bachman et al. (2019); Zhuang et al. (2019); Tian et al. (2019); Hénaff et al. (2019)). Meanwhile, the representations of different instances can be also distinguished between each other.

It is worth noting that these methods usually rely on image augmentations that are carefully designated to maintain their instance identities so that the augmentation of an instance can be accurately retrieved from a dictionary of instances. On the other hand, we believe stronger augmentations could expose novel patterns which can further improve the generalizability of learned representations and eventually close the gap with the fully supervised models. However, directly using stronger augmen-

tations in the contrastive learning could deteriorate the performance, because the induced distortions could ridiculously change the image structures and thus the transformed images cannot keep the identity of the original instances. Thus, additional efforts are needed for us to explore the role of the stronger augmentations to further boost the performance of self-supervised learning.

Thus we propose the CLSA (**C**ontrastive **L**earning with **S**tronger **A**ugmentations) framework to address this challenge. Instead of applying strongly augmented views to the contrastive loss, we propose to minimize the distribution divergence between the weakly and strongly augmented images over a representation bank to supervise the retrieval of stronger queries. This avoids an overoptimistic assumption that could overfit the strongly augmented queries containing distorted visual structures into the positive targets, while still being able to distinguish them from the negative samples by leveraging the distributions of weakly augmented counterparts. The learned representation will not only explore the novel patterns exposed by the stronger augmentations, but also inherits the knowledge about the relative similarities to the negative samples.

The experiments on various datasets demonstrate that the proposed framework can greatly boost the performance by learning from stronger augmentations. On the ImageNet linear evaluation protocol, we reach a record 76.2% top-1 accuracy with the standard ResNet-50 backbone, which is almost as high as 76.5% top-1 accuracy of the fully supervised model. Meanwhile, it also achieves the competitive performances on several downstream tasks. Among them is a top-1 accuracy of 93.6% on VOC07 with the linear classifier on the pretrained ResNet-50 compared to the previous record of 88.9% top-1 accuracy. For the COCO object detection, the $AP_S$ for small object detection has been improved to 24.4% from the previous best $AP_S$ of 20.8%. These results show that the CLSA can more effectively leverage stronger augmentations than the previous self-supervised methods on downstream tasks. We also conduct ablation study to show a naive application of stronger augmentations in the contrastive learning would degrade the performances.

## 2 RELATED WORK

Unsupervised and self-supervised learning methods have been widely studied to close the gap with supervised learning. These methods can be categorized into four different major aspects.

**Instance Discrimination and Contrastive Learning** Each image is considered as an individual class in an instance discrimination setting (Bojanowski & Joulin (2017); Dosovitskiy et al. (2015); Wu et al. (2018); Chen et al. (2020a); He et al. (2020)). It can be further formulated as contrastive learning (Hadsell et al. (2006)). In particular, Wu et al. (2018) built a memory bank that stores pre-computed representations from which positive examples are retrieved given some queries. Following this work, He et al. (2020) used a momentum update mechanism to maintain a long queue of negative examples for contrastive learning. Chen et al. (2020a) proposed a rich family of data augmentations on cropped images which has significantly boosted the classification accuracy. However, these methods failed to further improve the performance by naively applying stronger augmentations to minimize the contrastive loss, and this motivated the proposed work.

**Generative Methods** The generative methods typically adopt auto-encoders (Vincent et al. (2008); Kingma & Welling (2013)), and adversarial learning (Donahue et al. (2016); Donahue & Simonyan (2019)) to train an unsupervised representation. Usually, they focused on the pixel-wise information of images to distinguish images from different classes.

**Self-supervised Clustering** Data clustering (Asano et al. (2019); Caron et al. (2018; 2019; 2020); Yan et al. (2020)) can also be used to learn visual representations by assigning pseudo cluster labels to individual samples. DeepCluster (Caron et al. (2018)) generalized k-means by alternating between assigning pseudo-labels and updating networks. Recently, the SWAV (Caron et al. (2020)) is proposed to learn a cluster of prototypes as the negative examples for the contrastive learning. Combined with the multi-crops of training examples, the SWAV has achieved the state-of-the-art performance on ImageNet.

**Pretext Tasks** In addition to the contrastive learning, there exist many alternative methods using different pretext tasks (Agrawal et al. (2015); Qi et al. (2019); Doersch et al. (2015); Kim et al. (2018); Larsson et al. (2016); Zhang et al. (2019)) to train unsupervised deep networks. For example, Doersch et al. (2015) used the relative positions of two randomly sampled patches as the supervised signal. Agrawal et al. (2015); Zhang et al. (2019); Gidaris et al. (2018) adopted various geometric

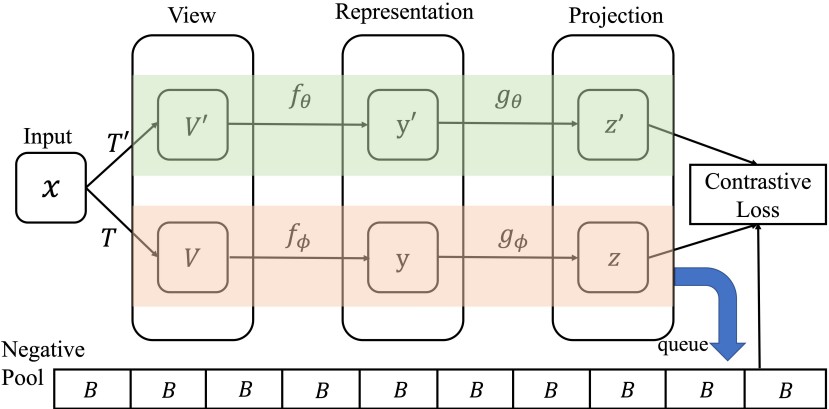

Figure 1: Contrastive instance learning framework

transformations on images and used the transformation parameters to train deep networks. For more details about these works, please refer to the survey by Jing & Tian (2020).

## 3 THE PROPOSED METHOD

In this section, we will first review the preliminary work on the contrastive learning, and discuss their limitations on applying stronger augmentations to explore the novel patterns of representations. Then we will present a new distributional divergence loss between weakly and strongly augmented images to self-train the representations over a representation bank consisting of both negative samples and positive targets. After that, the algorithm and the implementation details will also be explained.

### 3.1 PRELIMINARY METHODS AND LIMITATIONS

Contrastive learning (Hadsell et al. (2006)) is a popular self-supervised idea and made great success in recent years with the advance of computation and various image augmentations. In contrastive learning, each image is considered as an instance, so it's also known as instance learning.

Fig. 1 illustrates general framework of contrastive learning methods. For each image $x$ in batch $B$, we apply two transformations $T$ and $T'$ to obtain two different views $V$ and $V'$ of the same instance $x$. Then they go through a key encoder and a query encoder respectively, followed with MLP projection layers, resulting in two embedded representations $z$ and $z'$ to calculate the constrastive loss. The key factor of contrastive learning is the quality and the number of negative examples. To deal with those issues, various methods have been proposed, such as Memory Bank (Wu et al. (2018)), momentum encoder (He et al. (2020)), and online learning with bigger batch (Chen et al. (2020a)).

Specifically, the contrastive loss is developed to maximize the agreement of representations of different views of the same instance while minimizing the agreement with other negative samples. Hence, the contrastive loss for the different views of the same instance can be defined in Eq. (1).

$$\mathcal{L}_C = -\frac{1}{|B|} \sum_{i \in B} \log \frac{\exp(sim(z'_i, z_i)/\tau)}{\exp(sim(z'_i, z_i)/\tau) + \sum_{k=0}^{K} \exp(sim(z'_i, z_k)/\tau)} \quad (1)$$

with the cosine similarity

$$sim(z_i, z_j) = z_i^T z_j / (||z_i|| \cdot ||z_j||) \quad (2)$$

where $\mathcal{L}_C$ is the contrastive loss, $z_i$ and $z'_i$ are the projected representations of the different augmentations of the same sample $x_i$, the summation is taken over the samples $x_i$ in the current batch $B$, and $\tau$ is the temperature parameter set to 0.2. Also, the negative pool $Q = \{z_k | k = 1, \cdots, K\}$ shown in Fig. 1 is a queue of size $K$ storing the embedded features from the past batches in a FIFO fashion. This will keep the examples from the most recent batches in the queue while removing these obsoleted ones from it, which has been widely adopted in previous works (Chen et al. (2020b); He et al. (2020)).

As illustrated in Fig. 1, previous contrastive learning works use two transformations $T$ and $T'$ to generate two different views $V$ and $V'$, in which the transformations are carefully designated. Thus,

the two views are not transformed aggressively so that they can still be viewed as the same instance. However, directly adopting stronger transformations (e.g., with larger rotation angles, more aggressive colorjittering and cutout) in contrastive learning fails to further improve the performance or even deteriorate it for downstream tasks, which is not surprising. Stronger transformations could distort image structures and their perceptual patterns in the learned representation so that strongly augmented samples from the same instance cannot be viewed as keeping the same instance identity for training the underlying network. However, stronger augmentations can expose useful clues to the novel patterns that cannot be revealed from moderately augmented images. In supervised learning (Cubuk et al. (2018); Lim et al. (2019); Hataya et al. (2019); Cubuk et al. (2020)), data augmentation search have been widely studied and greatly boost the performance with the novel pattern exposed by strongly augmented images. The findings in RandAugment (Cubuk et al. (2020)) have verified that strongly augmented views can provide more clues even without an explicit augmentation policy. We believe learning the representations from these novel patterns will pave the last mile to close the gap with the fully supervised representations. Indeed, in semi-supervised learning and supervised learning (Cubuk et al. (2020); Qi et al. (2019); Wang et al. (2019)), more aggressive augmentations have been adopted and achieved extraordinary performances. For example, AET Qi et al. (2019) has adopted the parameters of augmentations as supervised signal to self-supervise the training of networks. All of these findings have inspired us to explore novel ways to utilize stronger transformations in self-supervised learning while avoiding deteriorated performances by naively using them in a contrastive model (Chen et al. (2020a)). All of these have inspired us to explore novel ways to utilize stronger transformations in self-supervised learning.

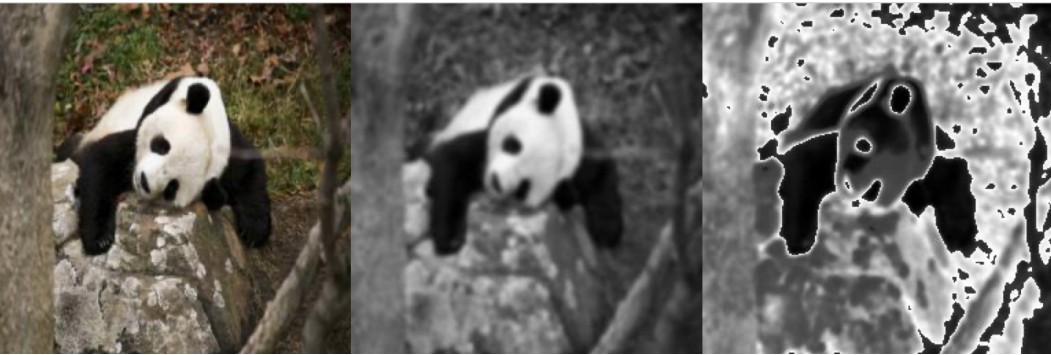

Figure 2: Comparison of the strongly weakly augmented images. The left is the original image, the middle is the weakly augmented image, and the right is the strongly augmented one with over-contrastive details.

However, it is not easy. As shown in Fig. 2, a strongly augmented image may look perceptually different from the original counterpart. Consequently, the representation of a strongly augmented image can be far apart from that of the weakly augmented one. Thus, naively using strongly augmented images in contrastive learning can be over-optimistic since the induced distortions could dramatically ruin their image structures. To this end, in Section 3.2, we instead proposed the Distributional Divergence Minimization (DDM) between weakly and strongly augmented images over a representation bank to avoid overfitting the representation of a strongly augmented image with that of the corresponding positive target.

## 3.2 Distributional Divergence Minimization between Weakly and Strongly Augmented Images

Due to the aforementioned limitation of stronger augmentations in contrastive learning, a direct retrieval of a strongly augmented query is infeasible to self-train deep networks. Fortunately, the distribution of relative similarities of a weakly augmented image from the same instance over the representation bank can provide useful information to bridge the gap. As explained below, it does not only avoid directly placing the representation of a strongly augmented image too closer to that of the positive target, but also allows it to explore the novel patterns of variations exposed by the strong augmentation.

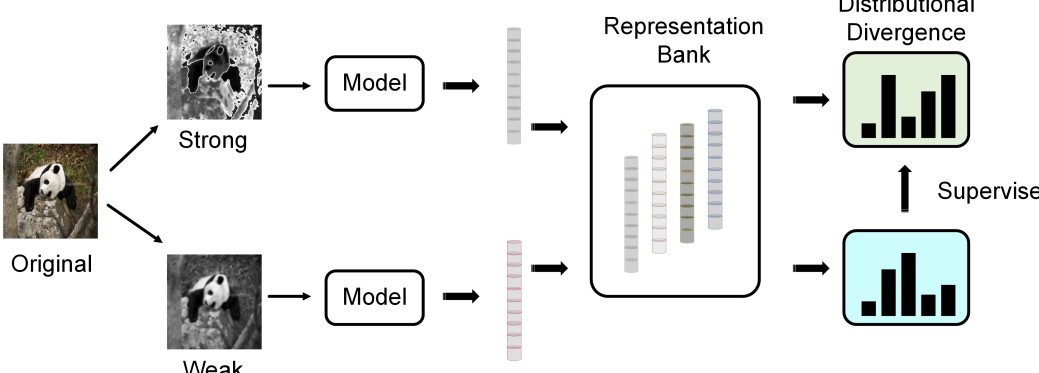

Figure 3: Diagram of distributional divergence minimization. Here the representation bank consists of $K$ stored features $z_k$ in the negative pool and online features $z_i$ from the key encoder. They will be used to calculate the conditional probability of current weakly and strongly augmented images.

Formally, as shown in Fig. 3, for an original image $x_i$ and the representation $z_i$ of the corresponding positive target, we applied a weaker and a stronger augmentation to obtain two separate views $V_i'$ and $V_i''$, and their embeddings $z_i'$ (weak representation) and $z_i''$ (strong representation). Given a pool $Q$ (shown in Fig. 1) of $K$ negative samples $\{z_k | k = 1, \cdots, K\}$ accumulated from the past iterations, we obtain a conditional distribution

$$p(z_k|z_i') = \frac{exp(sim(z_i', z_k)/\tau)}{exp(sim(z_i', z_i)/\tau) + \sum_{k=0}^{K} exp(sim(z_i', z_k)/\tau)} \tag{3}$$

which encodes the likelihood of the weaker query $z_i'$ being assigned to $z_k$. In the similar way, we can define the likelihood $p(z_i|z_i')$ of the query being assigned to the positive target $z_i$, as well as the likelihoods $p(z_i|z_i'')$ and $p(z_k|z_i'')$ for the stronger query $z_i''$. Here the representation bank consists of the negative pool $Q$ and the positive query targets $z_i$ from the current batch $B$.

Then, we propose to minimize the following distributional divergence between the weak and the strong queries such that

$$\mathcal{L}_D = \frac{1}{|B|} \sum_{i \in B} \left[ -\sum_{k=1}^{K} p(z_k|z_i') \log(p(z_k|z_i'')) - p(z_i|z_i') \log(p(z_i|z_i'')) \right] \tag{4}$$

By minimizing this divergence, we assume the learned representation $z_i''$ of the strongly augmented query should inherit the representation $z_i'$ of the weakly augmented one regarding not only its belief of the query being assigned to the corresponding positive target $z_i$, but also its relations with the negative samples $z_k$ in the representation bank through the conditional distribution $p(z_k|z_i')$.

This will prevent a direct overfitting of the strong query representation to the positive target as well as improve the generalization of the learned representation with additional clues from the other examples in the representation pool. In a more general sense, this extends the idea of knowledge distillation (Hinton et al. (2015)). However, we did not use the predicted labels by a teacher model to supervise the training of a student model as in the knowledge distillation. Instead, we used the distribution of the likelihoods of a weak query to supervise the retrieval of a strong query from a pool of representations.

### 3.3 IMPLEMENTATION DETAILS

Algorithm 1 gives the pseudo code to implement the proposed CLSA method. In the following, we will discuss the details about the applied strong and weak augmentations for distributional divergence minimization.

**Stronger Augmentations $S$** As explored in the previous works (Cubuk et al. (2018); Wang et al. (2019); Qi et al. (2019)), strong augmentations usually have two types: geometric and non-geometric augmentations. Specifically, we considered 14 types of augmentations: ShearX/Y, TranslateX/Y,

---

**Algorithm 1** Pseudo code for the proposed CLSA

---

**Input:** $f_\theta$, $f_\phi$: the query and the key encoder networks; $g_\theta$, $g_\phi$: the MLP projection layers for the query and the key; $Q$: a queue of representations of $K$ negative samples; $\alpha$: momentum decay for the key network; $\tau$: the temperature; $T$ and $T'$: weak augmentation; $S$: strong augmentation; $\beta$: the balancing coefficient.
1: Initialize the network $f_\phi = f_\theta$, $g_\phi = g_\theta$ and $Q$;
2: **for** $i$=1 to $steps$ **do**
3:     Fetch $x$ from the current batch $B$;
4:     $z'' = g_\theta(f_\theta(S(x)))$, $z' = g_\theta(f_\theta(T'(x)))$, $z = g_\phi(f_\phi(T(x)))$;
5:     Calculate the contrastive loss $\mathcal{L}_C$;                          ▷ see Eq. (1)
6:     Calculate distributional divergence loss $\mathcal{L}_D$;                ▷ see Eq. (4)
7:     Update the query network $f_\theta$ and $g_\theta$ with loss $\mathcal{L} = \mathcal{L}_C + \beta * \mathcal{L}_D$;
8:     Update the key network $f_\phi$ and $g_\phi$ with $\phi = \alpha\phi + (1-\alpha)\theta$;
9:     Update $Q$ with the representation $z$ output from the key network;
10: **end for**
**Output:** The representation encoder $f_\theta$.

---

Rotate, AutoContrast, Invert, Equalize, Solarize, Posterize, Contrast, Color, Brightness, Sharpness. The magnitude of each augmentation is significant enough to produce as strong augmentations as possible. More details are shown in Table 1. For example, the shear is drawn from a range of [-0.3,0.3], which results in aggressively transformed images that can be hard to retrieve given a counterpart target. In particular, to transform an image, we randomly select an augmentation from the above 14 types of transformations, and apply it to the image with a probability of 0.5. This process is repeated five times and that will strongly augment an image as the example shown in the right panel of Fig. 2.

Table 1: Various augmentations we applied in experiments to strongly augment training images.

| Operation | ShearX(Y) | TranslateX(Y) | Rotate | AutoContrast | Invert | Equalize |
|---|---|---|---|---|---|---|
| Mag Range | [-0.3,0.3] | [-0.3,0.3] | [-30,30] | 0 or 1 | 0 or 1 | 0 or 1 |
| Operation | Solarize | Posterize | Contrast | Color | Brightness | Sharpeness |
| Mag Range | [0,256] | [4,8] | [0.05,0.95] | [0.05,0.95] | [0.05,0.95] | [0.05,0.95] |

**Weaker Augmentations $T$** Weak augmentations are drawn by following most of existing contrastive learning methods in literature (Chen et al. (2020a;b); Caron et al. (2020); He et al. (2020)): an image is first cropped from an input image and resized to 224×224 pixels. Then random color jittering, Gaussian Blur, grayscale conversion, horizontal flip, channel-wise color normalization are sequentially applied to generate weakly augmented images with an example shown in the middle of Fig. 2.

**Technical Details** Similar to the previous works (He et al. (2020); Chen et al. (2020a); Caron et al. (2020)), we used the ResNet-50 (He et al. (2016)) as our encoder backbones $f_\theta$ and $f_\phi$ and a 2-layer MLP (2048-d hidden layer with the ReLU) as the projection head $g_\theta$ and $g_\phi$. The projected representation $z$ is first L2-normalized (Wu et al. (2018)) before calculating the cosine similarity. The temperature $\tau$ is set to 0.2, with a momentum smoothing factor $\alpha$ of 0.999 and a fixed balancing coefficient $\beta$ of 1.0. We set the size $K$ of the queue $Q$ to 65536 to store the negative examples used to compute the conditional distribution of weakly and strongly augmented queries and minimize their divergence.

## 4 EXPERIMENTS

### 4.1 TRAINING DETAILS

For the unsupervised pretraining on ImageNet with the CLSA, we used the SGD optimizer (Bottou (2010)) with an initial learning rate of 0.03, a weight decay of 0.0001 and a momentum of 0.9. We used cosine scheduler (Loshchilov & Hutter (2016)) to gradually decay the learning rate to 0. Usually, the batch size is set to 256. When multiple GPU cluster servers are used, the batch size will

Table 2: Top-1 accuracy under the linear evaluation on ImageNet with the ResNet-50 backbone. The left table compared methods trained over 200 epochs, and the right table compared the methods with various numbers of epochs.

| Method | Top 1 | Method | Top 1 |
|---|---|---|---|
| InstDisc (Wu et al. (2018)) | 54.0 | BigBiGAN (Donahue & Simonyan (2019)) | 56.6 |
| LocalAgg (Zhuang et al. (2019)) | 58.8 | SeLa-400epochs (Asano et al. (2019)) | 61.5 |
| MoCo (He et al. (2020)) | 60.8 | PIRL-800epochs (Misra & Maaten (2020)) | 63.6 |
| SimCLR (Chen et al. (2020a)) | 61.9 | CMC (Tian et al. (2019)) | 66.2 |
| CPC v2 (Hénaff et al. (2019)) | 63.8 | SimCLR-800epochs (Chen et al. (2020a)) | 70.0 |
| PCL (Li et al. (2020)) | 65.9 | MoCo v2-800epochs (Chen et al. (2020b)) | 71.1 |
| MoCo v2 (Chen et al. (2020b)) | 67.5 | InfoMin Aug-800epochs (Tian et al. (2020)) | 73.0 |
| InfoMin Aug (Tian et al. (2020)) | 70.1 | BYOL-1000epochs (Grill et al. (2020)) | 74.3 |
| SWAV (Caron et al. (2020)) | 72.7 | SWAV-800epochs (Caron et al. (2020)) | 75.3 |
| CLSA-Single | 69.4 | CLSA-Single-800epochs | 72.2 |
| CLSA-Multi | **73.3** | CLSA-Multi-800epochs | **76.2** |
| Supervised | 76.5 | Supervised | 76.5 |

be multiplied by the same number of servers by convention. Typically, the experiment with a single strong augmentation for each training image takes roughly 70 hours to finish on 8 V100 GPUs.

For the fine-tuning on ImageNet, we trained a linear classifier on top of the frozen feature vector (2048-D) upon the pretrained ResNet-50 with CLSA. This linear layer is trained for 100 epochs, with a learning rate of 10 without weight decay. We used the cosine learning rate decay and a batch size of 256.

For the transfer learning on the VOC dataset, we trained a linear classifier upon the pretrained Resnet-50 in the similar way for ImageNet – we trained 100 epochs with the SGD optimizer and a learning rate of 0.05, a momentum of 0.9 and no weight decay. The batch size is 256 without learning rate scheduler.

Finally, for object detection, we adopted the same protocol in (He et al. (2020)) to fine-tune the pretrained Resnet-50 backbone based on detectron2 (Wu et al. (2019)) for the sake of a fair and straight comparison with the other methods.

### 4.2 LINEAR CLASSIFICATION ON IMAGENET

For the linear evaluation on ImageNet, we trained the CLSA in two settings. In the first setting named CLSA-Single, we used a single stronger augmentation (see Table 1) that crops each training image to a smaller size of $96 \times 96$, which does not incur too much computing overhead in processing these smaller augmented images. In the second setting named CLSA-Multi, we adopted five different stronger augmentations that crop each image into various sizes: $224 \times 224, 192 \times 192, 160 \times 160, 128 \times 128$, and $96 \times 96$. The DDM loss in Eq. (4) is the sum over these multiple stronger augmentations. The similar multi-crop strategy has been adopted in contrastive learning literature before. For example, the SWAV (Caron et al. (2020)) reached the state-of-the-art top-1 accuracy by applying such multi-crop augmentations. To ensure a fair comparison with the SWAV, we chose five stronger augmentations such that the self-training with CLSA-Multi consumed the same computing time (i.e., 166 hours with a cluster of 8 V100 GPUs for 200 epochs of pre-training with a batch size of 256).

As shown in Table 2, we compared the performance with the other unsupervised methods. All the experiments are based on a pretrained ResNet-50 backbone that is fine-tuned with a linear classifier. The left table showed the performance of different methods pretrained over 200 epochs, and the right table reported models pretrained over more epochs.

First, under the same contrastive protocol, the CLSA-Single has a higher $69.4\%$ top-1 accuracy than both MoCo v2 (67.5%) and SimCLR (61.9%) with 200 epochs training. With multiple stronger augmentations, the CLSA-Multi outperforms the State-of-the-art SWAV model using multi-crops of training images over 200 epochs 73.3% vs. 72.7%. Moreover, as shown in the right table, the CLSA-

Single outperforms MoCo v2 and SimCLR with the same training epochs. It is also noteworthy that the CLSA-Multi achieves almost the same top-1 accuracy as that of the fully supervised network (76.2% vs. 76.5%).

## 4.3 TRANSFER LEARNING RESULTS ON DOWNSTREAM TASKS

Table 3: Transfer learning results on various downstream tasks.

| Measurement | Classification VOC07 Accuracy | Object Detection VOC07+12 AP$_{50}$ | COCO AP | AP$_S$ |
|---|---|---|---|---|
| RotNet (Gidaris et al. (2018)) | 64.6 | - | - | - |
| NPID++ (Wu et al. (2018)) | 76.6 | 79.1 | - | - |
| MoCo (He et al. (2020)) | 79.8 | 81.5 | - | - |
| PIRL (Misra & Maaten (2020)) | 81.1 | 80.7 | - | - |
| PCL (Li et al. (2020)) | 84.0 | - | - | - |
| BoWNet (Gidaris et al. (2020)) | 79.3 | 81.3 | - | - |
| SimCLR (Chen et al. (2020a)) | 86.4 | - | - | - |
| MoCov2 (Chen et al. (2020b)) | 87.1 | 82.5 | 42.0 | 20.8 |
| SWAV (Caron et al. (2020)) | 88.9 | 82.6 | 42.1 | 19.7 |
| CLSA | **93.6** | **83.2** | **42.3** | **24.4** |
| Supervised | 87.5 | 81.3 | 40.8 | 20.1 |

We test the generalizability of the ResNet-50 pre-trained on ImageNet to several downstream tasks. Specifically, we focused on two tasks: cross-dataset image classification and object detection. The pre-trained ResNet-50 was frozon, and we fine-tuned the linear classifier on the VOC07trainval (Everingham et al. (2010)) and tested it on the VOC07test. For object detection, we evaluated the pre-trained network on two datasets using the detectron2 (Wu et al. (2019)) used in the previous methods He et al. (2020); Chen et al. (2020a). On the VOC dataset (Everingham et al. (2010)), we trained the detection head with VOC07+12 trainval dataset and tested on VOC07 test dataset. On the COCO dataset (Lin et al. (2014)), we fine-tuned the network on the train2017 set with 118k images and evaluate on the val2017. For the sake of a fair comparison, the object detection tasks are completed by detectron2 (He et al. (2017)) based on the pretrained ResNet-50.

As shown in Table 3, the performances on both tasks are much better than the supervised model trained on ImageNet. This suggests that the proposed method has better generalization ability in downstream tasks. The pre-trained network on ImageNet by the CLSA outperformed the compared models after being fine-tuned on different datasets. Among them is a top-1 accuracy of 93.6% on the VOC07 with the linear classifier on the pretrained ResNet-50 in comparison with the previous record of 88.9% top-1 accuracy by the SWAV. On the COCO dataset, the $AP_S$ for small object detection has been significantly improved to 24.4% from the previously best $AP_S$ of 20.8%. As well known, it is much challenging to detect small objects on the COCO dataset. Thus, the better performance of the CLSA could be attributed to the ability of involving the stronger augmentations that result in many small objects to pretrain the network.

## 4.4 ABLATION STUDY

Table 4: Ablation study of the CLSA on ImageNet with 200 epochs of pre-training.

| Model | Top-1 Accuracy |
|---|---|
| MoCo V2 | 67.5 |
| MoCo V2 with Strong query | 67.7 |
| MoCo V2 with Strong query & Strong key | 67.0 |
| CLSA-Single with contrastive loss | 68.0 |
| CLSA-Single | 69.4 |

In the ablation study shown in Table 4, we studied the role of the proposed DDM loss in the CLSA. First, we naively used the stronger augmentation applied in the CLSA-Single as the query and/or

the key in the MoCo V2. Both results (Strong query and Strong query & Strong key) showed the performance can not be improved or even degraded. Second, we replaced the DDM loss in the CLSA-Single with the contrastive loss, and we found it can only achieved a top-1 accuracy of 68.0% compared to that of 69.4% with the DDM loss. Both studies showed that the proposed CLSA and its DDM loss help us learn from stronger augmentations by avoiding the performance degeneration that would be incurred by the distortions of augmented images.

## 5 CONCLUSION

In this paper, we present CLSA, a novel method that can utilize the distributional divergence to learn the information from strongly augmented images. The proposed method outperforms the state-of-the-art methods on all the datasets and achieved almost same performance compared to supervised ImageNet network. Meanwhile, it outperforms the previous supervised and self-supervised methods on downstream tasks, which suggests CLSA learned more reliable and fine-grained features that can contribute to the development of other areas.

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
