# OpenReview forum: "Contrastive Learning with Stronger Augmentations"
_ICLR.cc/2021/Conference — Reject_

### Official Review · AnonReviewer2 · 2020-10-25
**Official Blind Review #2**

**Rating:** 6
**Confidence:** 4

**Review:**

Summary:\
This work investigate the recent popular direction of unsupervised representation learning using contrastive loss between augmented images. Authors propose to minimize the divergence between the distributions of strongly augmented vs. weakly augmented images. The method reaches competitive performance in recognition and object detection.
-----
+Strengths\
+The main idea is well motivated: that strong augmentation reveal useful cues in visual representation learning but has not been successfully exploited in unsupervised learning.\
+The proposed solution is novel within contrastive learning to my best knowledge.\
+Results are extremely strong.
-----
-Concerns\
-The divergence between the two conditional distributions can be a moving target since they are trained jointly. It is not clear if this is will result in stable learning for the unsupervised setting, and what effect that may have on the performance and quality of the representations.\
-Evaluation only focus on final result and lacks analysis of the proposed method, especially when compared to recent paper of similar nature published in top conferences. For example, strong augmentation is a focus of this paper, but there are no ablation regarding the augmentations. Is the performance sensitive to the choice of strong augmentation?\
-The paper could also use some more theoretical analysis to address some of the weaknesses stated above.
-----
Recommendation\
I like the proposed idea. It is novel and interesting and seems to achieve good results. However the lack of both theoretical and empirical analysis beyond results on performance raises many questions. As a result I am on the fence but leaning towards accept.

---

> ### Author Response · Authors · 2020-11-15
> **Response**
>
> Thanks a lot for your reviewing and kind suggestions!
>
> 1. The target conditional distribution of a weakly augmented view is computed over the pool of negative examples that have already been stabilized with momentum updates. Thus, this conditional distribution is relatively stable, and can be used to supervise the conditional distribution of the associated strongly augmented views. In our experiments, we also found the DDM loss can stably converge at the same rate of the conventional contrastive loss, with the learned representation demonstrating competitive performance on downstream tasks.
> 2. In one of the previous works, InfoMin [1] studied the impact of different augmentations on the overall performance, which showed that the contrastive learning can be sensitive to the augmentations parameters. However, for the augmentations we used in experiments, we avoid this by randomly sampling various magnitudes of augmentations from a much broad range, including those resulting in much stronger augmentations. We did not carefully tune the strength of the resultant stronger augmentations, and we found the performance is insensitive to these sampled augmentations.  This is unlike the method such as InfoMin [1] and AutoAugment [2] that tunes or searches for a specific level of augmentation strength for an optimal performance over a training/validation set. This makes the proposed CLSA more generalizable in adopting a wide range of stronger augmentations without relying on tedious tuning for a suitable strength.
> 3. Thank you for the suggestion! Our main idea is to use the conditional distribution of weakly augmented views over negative examples to supervise that of strongly augmented views. This avoids a direct engagement of stronger augmentations in the minimization of contrastive loss that could deteriorate the performance as shown in our ablation study (see Table 4).  The similar phenomenon has been observed in knowledge distillation, where directly using noisy labels to supervise the network training is worse than using the predicted labels from another network.   We will try to develop some theory to analyze it.
>
>
> [1] Tian, Yonglong, et al. "What makes for good views for contrastive learning." arXiv preprint arXiv:2005.10243 (2020).
> [2] Cubuk, Ekin D., et al. "Autoaugment: Learning augmentation strategies from data." Proceedings of the IEEE conference on computer vision and pattern recognition. 2019.

---

### Official Review · AnonReviewer3 · 2020-10-27
**Official Blind Review #3**

**Rating:** 6
**Confidence:** 3

**Review:**

This paper presents a method to incorporate stronger augmentations into the visual representation contrastive learning framework. Specifically, three correlated views of an image are first generated by using two weak and one strong augmentation operations on the same image. Then, the networks are trained to maximize the agreement between the two weak views and also to minimize the distribution divergence between a weak view and the strong view. The method is evaluated on several visual tasks including classification,  transfer learning, and object detection, with the standard evaluation protocol for self-supervised learning, and the results are promising.

Pros:

1. This paper is well-structured and easy-to-follow.
2. The idea of utilizing strong augmentations for contrastive learning is interesting and novel to me, and the results are promising.
3. The proposed framework seems general which might be easily incorporated into the existing contrastive learning frameworks.

Cons:

1. The motivation about using stronger augmentations is not well justified. Specifically, the authors propose to use stronger augmentations based on two reasons: (1) stronger augmentations can expose some novel useful patterns; (2) the effectiveness of stronger augmentations is proved in the semi-supervised learning and supervised learning field. However, no related papers are provided to support the first point, while the papers (Cubuk et al. (2018)); Qi et al. (2019); Wang et al. (2019)) that are cited to support the second point do not explicitly make relevant conclusions. (Chen et al. (2020a)) even demonstrate that when training supervised models, stronger color augmentation hurts their performance. I would like to see a more comprehensive review of related works to clarify the motivation.

2. In addition, some important ablation studies are missing in the experiment. E.g.,  how does the performance change as the magnitude or usage times of stronger augmentations changes?

3. The proposed DDM loss seems general for different contrastive learning frameworks. I would like to see if it still works when applied to other frameworks, e.g., SimCLR, InfoMin?

Overall, given the novelty and strong results of the proposed framework, I remain positive towards this paper. I will be happy to increase my rating if my concerns are addressed in the rebuttal period.

---

> ### Author Response · Authors · 2020-11-15
> **Response**
>
> Thank you very much for your kind review and suggestion!
>
> 1. We have polished our motivation in the revised paper. Below is the revised motivation.
> “However, stronger augmentations can expose useful clues to the novel patterns that cannot be revealed from moderately augmented images. In supervised learning [1,2,3,4], data augmentation search have been widely studied and greatly boost the performance with the novel pattern exposed by strongly augmented images. The findings in RandAugment [4] have verified that strongly augmented views can provide more clues even without an explicit augmentation policy.  We believe learning the representations from these novel patterns will pave the last mile to close the gap with the fully supervised representations.  Indeed, in semi-supervised learning and supervised learning [4,5,6], more aggressive augmentations have been adopted and achieved extraordinary performances.  For example, AET [5] has adopted the parameters of augmentations as supervised signal to self-supervise the training of networks. All of these findings have inspired us to explore novel ways to utilize stronger transformations in self-supervised learning while avoiding deteriorated performances by naively using them in a contrastive model [8].”
> 2. We do not finetune the augmentation magnitudes in our experiments. The strength of our augmentations is randomly sampled from a broad range, and we randomly pick one of the 12 augmentation types each time with the same probability. This differs from as InfoMin [7] and AutoAugment [1] searching for an optimal level of augmentation strength over a training/validation set. The experiment results show that the proposed CLSA is insensitive to these randomly sampled augmentations, and it successfully explores the stronger augmentations to improve the performance of learned representations in downstream tasks.
> 3. We agree that the proposed DDM loss could also work well in the other contrastive learning framework.  However, due to the limited time and the rebuttal policy, we could not do a very exhaustive investigation into such possibility. But we will report these results in our extended research.
>
>
> [1] Cubuk, Ekin D., et al. "Autoaugment: Learning augmentation strategies from data." Proceedings of the IEEE conference on computer vision and pattern recognition. 2019.
> [2] Lim, Sungbin, et al. "Fast autoaugment." Advances in Neural Information Processing Systems. 2019.
> [3] Hataya, Ryuichiro, et al. "Faster autoaugment: Learning augmentation strategies using backpropagation." arXiv preprint arXiv:1911.06987 (2019).
> [4] Cubuk, Ekin D., et al. "Randaugment: Practical automated data augmentation with a reduced search space." Proceedings of the IEEE/CVF Conference on Computer Vision and Pattern Recognition Workshops. 2020.
> [5] Zhang, Liheng, et al. "Aet vs. aed: Unsupervised representation learning by auto-encoding transformations rather than data." Proceedings of the IEEE Conference on Computer Vision and Pattern Recognition. 2019.
> [6] Wang, Xiao, et al. "Enaet: Self-trained ensemble autoencoding transformations for semi-supervised learning." arXiv preprint arXiv:1911.09265 (2019).
> [7] Tian, Yonglong, et al. "What makes for good views for contrastive learning." arXiv preprint arXiv:2005.10243 (2020).
> [8] Chen, Ting, et al. "A simple framework for contrastive learning of visual representations." arXiv preprint arXiv:2002.05709 (2020).

---

### Official Review · AnonReviewer4 · 2020-10-28
**CONTRASTIVE LEARNING WITH STRONGER AUGMENTATIONS**

**Rating:** 7
**Confidence:** 4

**Review:**


This paper proposes the better utilization of strong data augmentations for contrastive loss functions in unsupervised learning. In Moco set up, typically, weaker augmentations such as color jittering, cropping is applied to construct positive pairs from the same image. In this study, by proposing a modified objective, the authors leverage stronger data augmentations to construct more challenging positives and negatives pairs to improve the quality of the representations. The paper delivers a novel objective together with leveraging existing strong augmentations to improve downstream performance. The authors can find my questions/concerns listed below.

1. The paper is overall well-written, however, it is disappointing to see many typos grammar mistakes throughout the paper. Some examples are in "Thus we proposed the CLSA (Contrastive Learning with Stronger Augementations)", "to train an unsupervised representation", "The contrastive learning (Hadsell et al. (2006)) is a popular self-supervised idea".

2. In section 3.1, the authors mention that the keys in the memory bank is managed with first in first out method. Is it not supposed to be first in last out? I would like to see some clarification on this.

3.  The numerator in Equation 3 should be z_i' vs. z_i not z_k.

4. The authors claim that in He et al. an input image is resized and cropped to 224×224 pixels. It should be "an image is first cropped from an input image and resized to 224x224 pixels."

5. In the experiments section, the authors list other methods including MoCo, SimCLR, MoCo-v2, BYOL and compare to what they propose. As a baseline, it would be nice to directly use the stronger augmentations in MoCo-v2 objective and perform comparison to their method. Throughout the paper, the authors claim that strong augmentations hurt the learned representations due to distorted images. It would be meaningful to show this experimentally as well.

6. The authors explain that they choose a strong transformation randomly from the given 14 transformations and repeat it 5 times to strongly augment an image. Is the sampling done without replacement? In other words, do the authors choose 5 unique transformations with the corresponding magnitude and apply those transformations to a single image?

7. I like how the authors point the similarity of their objective to knowledge distillation. In this case, strong augmentations are assigned probability of being a positive pair from the positive pair constructed with weak augmentations. It helps to understand the full picture for the proposed method.

8. Finally, I think the figure 3 is confusing rather than being helpful. Both weak and strong augmentations go to the memory bank and it looks like two distributions come out of nowhere in the figure. It would be more clear to point out that there is distribution of the representations from the strong augmentations and weak augmentations and they supervise the assignment for strong augmentations given predictions on the weak augmentations.

---

> ### Author Response · Authors · 2020-11-15
> **Response**
>
> Thank you so much for your careful review and kind suggestions!
>
> 1. We have polished those typos in the revised version.
>
> 2. In self-supervised learning like the MoCo, a queue of negative examples are maintained in a First-In-First-Out fashion. This will keep the examples from the most recent batches in the queue while removing these obsoleted ones from it.  We will clarify this in the revised submission.
>
> 3. In Eq. (3), we aim to compute the conditional distribution of a given image z_i’ over the negative examples (z_k, k=1,…,K). thus, it is indeed z_i’ vs. z_k not z_i’’. As explained in the manuscript, we use the conditional distribution of weakly augmented views to supervise that of strongly augmented views.
>
> 4. We have updated this description in the new revision.
>
> 5. We had conducted such an ablation study of MoCo V2 in Table 4. If directly applying strongly augmented views in MoCo V2, the result showed that it would hurt the performance.
>
> 6. We did it without replacement. Because the strength of augmentations are sampled from a continuous range, the chance of sampling two identical augmentations is almost zero.
>
> 7. We do think it can be regarded as a form of knowledge distillation from the conditional distributions over the memory bank in the self-supervise learning. We are happy that helps you to understand the method better.
>
> 8. We plot the Representation bank to illustrate that the distribution is calculated over the given representation bank shown in Eq.(3) and Eq.(4). We have added more descriptions for the figure to make it easier to understand.
>
> [1] Chen, Xinlei, et al. "Improved baselines with momentum contrastive learning." arXiv preprint arXiv:2003.04297 (2020).

---

> > ### Comment · AnonReviewer4 · 2020-11-23
> > **CONTRASTIVE LEARNING WITH STRONGER AUGMENTATIONS**
> >
> > Thanks for your replies to my question.
> >
> > My only question/concern remaining with this paper is the computational complexity comparison of CLSA-single to MoCo-v2. It seems that when training them for the same number of hours, the improvement is marginal, 0.8%. Can you also comment on the computational complexity of CLSA-multiple and compare it to MoCo-v2?

---

> > > ### Author Response · Authors · 2020-11-24
> > > **Thanks for your comments!**
> > >
> > > Thanks a lot for your kind comments!
> > >
> > > First, the CLSA indeed solidly beats the SOTA method SWAV that must consume much more time. For example, SWAV uses 173 hours (8 gpus) to train over 200 epochs to beat the MoCo V2, which is around 3 times of training time. Those methods cannot beat MoCo V2 if trained with the same amount of time. To our best knowledge, we are the (if not the only) method that successfully beats MoCo V2 using the same time. Moreover, with slightly increased training time for 200 epochs training, we can improve the MoCo V2 by over 2% in top-1 accuracy.
> > >
> > > Furthermore, for a fair comparison with the SOTA SWAV model based on multi-crop augmentations, the CLSA-Multi also performs better after being trained with less time, 1328 gpu hours (CLSA-Multi) vs. 1384 gpu hours (SWAV). This shows that the CLSA in both single-crop and multi-crop versions outperforms the SOTA method when they are compared with the same amount of training time.

---

### Official Review · AnonReviewer1 · 2020-11-03
**The experimental evaluations are not convincing**

**Rating:** 4
**Confidence:** 4

**Review:**

This paper focuses on designing more effective ways for contrastive learning. The author claims that stronger augmentations are beneficial for better representation learning. Different from directly applying the stronger augmentations to minimize the contrastive loss, the author proposes to minimize the distribution divergence between the weakly and strongly augmented images. The experimental evaluations are conducted on ImageNet classification and related downstream tasks, and the results are promising.

Clarity:
1. The method is very simple and straightforward. My main concern is the experimental comparisons. As we all know, contrastive learning algorithms like MOCO and SimCLR benefits from longer training epochs a lot (for example, training with 800 epochs is much better than with 400 epochs). Thus I think the comparisons in Table 2 are not convincing. From algorithm 1, we can find that the equivalent batch size of the proposed CLSA method is two times as classical MOCO method. Thus I would prefer to check the results of CLSA at epochs 100 and 400 for fair comparisons.
2. What is the value of the balancing coefficient? It would be nice if some ablation results are provided.

---

> ### Author Response · Authors · 2020-11-15
> **Response**
>
> Thanks a lot for your time for reviewing this paper and your constructive comments!
>
> 1. As we mentioned in the submission, the equivalent batch size is NOT twice as the compared method like MoCo.  We adopted a much smaller 96*96 cropped images for the strongly augmented view to calculate the proposed DDM loss. Accordingly, the running time of CLSA-single for 200 epochs is only 1.3 times (70 hours) that of the MoCo v2 (53 hours).  With 150 epochs, the CLSA-single only needs 52.5 hours, which is the same as MoCo v2 running for 200 epochs. In this case, the accuracy of CLSA-single is still higher than MoCo V2 with 68.3% vs 67.5%. Due to the time limit, we did not run the CLSA-single for 600 epochs that has a comparable running time for MoCo v2 over 800 epochs. But its accuracy should still be higher than MoCo v2.
>
> 2. In experiments, we simply set the balancing coefficient to 1.0, which is the most natural choice without any tuning over it.  Although adjusting this hyper-parameter could further improve the performance, the proposed CLSA has already achieved very competitive results compared with the other models.

---

### Public Comment · ~Tianyuan_Zhang2 · 2020-11-10
**Unofficial implementation**

I am implementing this algorithm using pytorch at https://github.com/a1600012888/clsa_pytorch . The implementation is most adopted from the offical moco's implementation.

---

> ### Author Response · Authors · 2020-11-15
> **Thanks for your implementation!**
>
> Thanks for your attention to this work and your time for implementation! We will release the code upon acceptance.

---

### Decision · Program_Chairs · 2021-01-07
**Final Decision**

**Decision:**

Reject

**Comment:**

This paper improves MoCo-based contrastive learning frameworks by enabling stronger views via an additional divergence loss to the standard (weaker) views. Three reviewers suggested acceptance, and one did rejection. Positive reviewers found the proposed method is novel and shows promising empirical results. However, as pointed out by the negative reviewer, the paper should have clarified about computational overheads of the method compared to the baseline (MoCo) in several aspects, e.g., their effective batch sizes or training costs, for the readers’ better understanding. As the concern was not fully resolved during the discussion phase, AC is a bit toward for reject. AC thinks the paper would be stronger if the authors include more ablations (and the respective discussions) regarding this point, e.g., CLSA-multi (and -single) vs. MoCo-v2 under the same training time, both at early epochs (~200; as reported in the author response) and longer epochs (after convergence; ~1000 and even more).